# Development of a Novel Anti-CD44 Variant 7/8 Monoclonal Antibody, C_44_Mab-34, for Multiple Applications against Oral Carcinomas

**DOI:** 10.3390/biomedicines11041099

**Published:** 2023-04-05

**Authors:** Hiroyuki Suzuki, Kazuki Ozawa, Tomohiro Tanaka, Mika K. Kaneko, Yukinari Kato

**Affiliations:** Department of Antibody Drug Development, Tohoku University Graduate School of Medicine, 2-1 Seiryo-machi, Aoba-ku, Sendai 980-8575, Miyagi, Japan

**Keywords:** CD44, CD44 variant 7/8, monoclonal antibody, flow cytometry, immunohistochemistry

## Abstract

Cluster of differentiation 44 (CD44) has been investigated as a cancer stem cell (CSC) marker as it plays critical roles in tumor malignant progression. The splicing variants are overexpressed in many carcinomas, especially squamous cell carcinomas, and play critical roles in the promotion of tumor metastasis, the acquisition of CSC properties, and resistance to treatments. Therefore, each CD44 variant (CD44v) function and distribution in carcinomas should be clarified for the establishment of novel tumor diagnosis and therapy. In this study, we immunized mouse with a CD44 variant (CD44v3–10) ectodomain and established various anti-CD44 monoclonal antibodies (mAbs). One of the established clones (C_44_Mab-34; IgG_1_, kappa) recognized a peptide that covers both variant 7- and variant 8-encoded regions, indicating that C_44_Mab-34 is a specific mAb for CD44v7/8. Moreover, C_44_Mab-34 reacted with CD44v3–10-overexpressed Chinese hamster ovary-K1 (CHO) cells or the oral squamous cell carcinoma (OSCC) cell line (HSC-3) by flow cytometry. The apparent *K*_D_ of C_44_Mab-34 for CHO/CD44v3–10 and HSC-3 was 1.4 × 10^−9^ and 3.2 × 10^−9^ M, respectively. C_44_Mab-34 could detect CD44v3–10 in Western blotting and stained the formalin-fixed paraffin-embedded OSCC in immunohistochemistry. These results indicate that C_44_Mab-34 is useful for detecting CD44v7/8 in various applications and is expected to be useful in the application of OSCC diagnosis and therapy.

## 1. Introduction

Head and neck cancers mainly arise from the oral cavity, pharynx, larynx, and nasal cavity. These tumors exhibit strong associations with smoking tobacco products, alcohol, and infection with human papillomavirus (HPV) types 16 and 18 [1]. The estimated number of new cases in the oral cavity and pharynx in the United States increased from 35,310 in 2008 to 54,540 in 2023 due to rising HPV-positive cases [2,3,4]. Mortality rates continue to increase for the oral cavity cancers associated with HPV infection (cancers of the tongue, tonsil, and oropharynx) by about 2% per year in men and 1% per year in women [2].

Although many different histologies exist in head and neck cancers, head and neck squamous cell carcinoma (HNSCC) is the common type. The treatment options for HNSCC include surgery, chemo-radiation, molecular targeted therapy, immunotherapy, or a combination of these modalities [5]. Despite the development in cancer treatment, metastasis and drug resistance remain the main causes of death [6]. Although survival can be improved, the impairment due to surgery and the toxicities of treatments deteriorate the patient’s quality of life. Thus, the 5-year survival rate remains stagnant at approximately 50% [1].

Cancer stem cells (CSCs) play critical roles in tumor development through their important properties, including self-renewal, resistance to therapy, and tumor metastasis [7,8,9]. Studies have reported the importance of CSCs in HNSCC development [10] and regulation by both intrinsic and extrinsic mechanisms in the tumor microenvironment [11]. Several cell surface receptors and intracellular proteins have been reported as applicable CSC markers in HNSCC [12,13]. Among them, cluster of differentiation 44 (CD44) is one of the important CSC markers in solid tumors, and it was first applied to study HNSCC-derived CSCs [14]. Notably, CD44-high CSCs from HNSCC tumors exhibited the properties of epithelial to mesenchymal transition, including elevated migration, invasiveness, and stemness [15]. Furthermore, CD44-high cells could form lung metastases in immunodeficient mice, in contrast to CD44-low cells, which failed to exhibit a similar metastatic proliferation of cancer cells [16]. Therefore, specific monoclonal antibodies (mAbs) against CD44 are required for the isolation of CD44-high CSCs and the analysis of their properties in detail.

CD44 is a multifunctional transmembrane protein that binds to the extracellular matrix, including hyaluronic acid (HA) [17]. Human CD44 has 19 exons, 10 of which are constant or present in all variants and make up the standard form of CD44 (CD44s). The CD44 variants (CD44v) are produced by alternative splicing and consist of the 10 constant exons in any combination with the remaining nine variant exons [18]. The CD44 isoforms have both overlapping and unique roles. Both CD44s and CD44v (pan-CD44) possess HA-binding motifs that promote interaction with the microenvironment and facilitate the activation of various signaling pathways [19].

Overexpression of CD44v has been observed in many types of carcinomas and is considered a promising target for tumor diagnosis and therapy [20,21]. There is growing evidence that CD44v plays important roles in the promotion of tumor metastasis, the acquisition of CSC properties [22], and resistance to chemotherapy and radiotherapy [23,24]. Several variant exon-encoded regions have been reported to promote tumorigenesis through their interacting proteins. The v3-encoded region is modified by heparan sulfate, which promotes the recruitment of heparin-binding growth factors such as fibroblast growth factors. Thus, the v3-encoded region functions as a co-receptor of receptor tyrosine kinases [25]. Furthermore, the v6-encoded region has been reported to be essential for the activation of c-MET through the formation of ternary complexes with HGF [26]. Moreover, the v8–10-encoded region mediates oxidative stress resistance through the regulation of intracellular redox states. [27]. Therefore, CD44v-specific mAbs are required not only for the understanding of each variant function but also for CD44v-targeting tumor diagnosis and therapy. However, the function and distribution of the variant-encoded region in tumors have not been fully understood.

Our group has developed the Cell-Based Immunization and Screening (CBIS) method and established a novel anti-pan-CD44 mAb, C_44_Mab-5 (IgG_1_, kappa) [28]. We also established another anti-pan-CD44 mAb, C_44_Mab-46 (IgG_1_, kappa) [29], using the immunization of CD44v3–10 ectodomain (CD44ec). We determined the epitopes of C_44_Mab-5 and C_44_Mab-46 in the standard exons (1 to 5)-encoding sequences [30,31,32]. We further showed that both C_44_Mab-5 and C_44_Mab-46 are available for flow cytometry, Western blot, and immunohistochemistry in oral SCC (OSCC) [28] and esophageal SCC [29]. Furthermore, we have also investigated the antitumor effects using recombinant C_44_Mab-5 in mouse xenograft models of oral OSCC [33]. We converted the mouse IgG_1_ subclass antibody (C_44_Mab-5) into an IgG_2a_ subclass antibody (5-mG_2a_) and further produced a defucosylated version (5-mG_2a_-f) using FUT8-deficient ExpiCHO-S (BINDS-09) cells. The 5-mG_2a_-f showed moderate in vitro ADCC and CDC activities against HSC-2 and SAS OSCC cell lines. Furthermore, the 5-mG_2a_-f significantly suppressed the xenograft growth of HSC-2 and SAS compared to control mouse IgG [33]. Here, we have developed a novel anti-CD44v7/8 mAb, C_44_Mab-34 (IgG_1_, kappa), and examined its applications to flow cytometry, Western blotting, and immunohistochemical analyses.

## 2. Materials and Methods

### 2.1. Cell Lines

Chinese hamster ovary (CHO)-K1, a human glioblastoma cell line (LN229), and mouse multiple myeloma P3X63Ag8U.1 (P3U1) cell lines were obtained from the American Type Culture Collection (ATCC, Manassas, VA, USA). The human OSCC cell line, HSC-3, was obtained from the Japanese Collection of Research Bioresources (Osaka, Japan). CHO-K1 and P3U1 were cultured in Roswell Park Memorial Institute (RPMI)-1640 medium (Nacalai Tesque, Inc., Kyoto, Japan), supplemented with 100 U/mL penicillin, 100 μg/mL streptomycin, 0.25 μg/mL amphotericin B (Nacalai Tesque, Inc.), and 10% heat-inactivated fetal bovine serum (FBS; Thermo Fisher Scientific, Inc., Waltham, MA, USA).

LN229 and HSC-3 were cultured in Dulbecco’s modified Eagle medium (DMEM) (Nacalai Tesque, Inc.), supplemented with 10% (*v*/*v*) FBS, 100 U/mL of penicillin (Nacalai Tesque, Inc.), 100 μg/mL streptomycin (Nacalai Tesque, Inc.), and 0.25 μg/mL amphotericin B (Nacalai Tesque, Inc.). LN229/CD44ec was cultured in the presence of 0.5 mg/mL of G418 (Nacalai Tesque, Inc.).

All the cells were grown in a humidified incubator at 37 °C with 5% CO_2_.

### 2.2. Plasmid Construction and Establishment of Stable Transfectants

Human CD44v3–10 open reading frame (ORF) was obtained from the RIKEN BRC through the National Bio-Resource Project of the MEXT, Japan. CD44s cDNA was amplified using a HotStar HiFidelity Polymerase Kit (Qiagen Inc., Hilden, Germany) using LN229 cDNA as a template. The CD44s and CD44v3–10 ORFs were subcloned into a pCAG-Ble-ssPA16 vector possessing signal sequence and N-terminal PA16 tag (GLEGGVAMPGAEDDVV) [28,34,35,36,37], which is detected by NZ-1, which was originally developed as an anti-human podoplanin mAb [38,39,40,41,42,43,44,45,46,47,48,49,50,51,52,53].

CHO/CD44s and CHO/CD44v3–10 were established by transfecting the plasmids into CHO-K1 cells using a Neon transfection system (Thermo Fisher Scientific, Inc.). CD44ec-secreting LN229 (LN229/CD44ec) was established by transfecting pCAG-Neo/PA-CD44ec-RAP-MAP into LN229 cells using the Neon transfection system. The amino acid sequences of the tag system in this study were as follows: PA tag [43,47,51], 12 amino acids (GVAMPGAEDDVV); RAP tag [54,55], 12 amino acids (DMVNPGLEDRIE); and MAP tag [56,57], 12 amino acids (GDGMVPPGIEDK).

### 2.3. Purification of CD44ec

The purification of CD44ec from the culture supernatant of LN229/CD44ec was performed using an anti-RAP tag mAb (clone PMab-2) and a RAP peptide (GDDMVNPGLEDRIE) [54,55]. The culture supernatant (5 L) was passed through a 2 mL bed volume of PMab-2-sepharose, and the process was repeated three times. After washing the beads with 100 mL of phosphate-buffered saline (PBS, Nacalai Tesque, Inc.), CD44ec was eluted with 0.1 mg/mL of a RAP peptide in a step-wise manner (2 mL × 10). The purity of CD44ec was determined by Coomassie Brilliant Blue (CBB) staining using the Bio-Safe CBB G-250 Stain (Bio-Rad Laboratories, Inc., Berkeley, CA, USA) (Appendix A).

### 2.4. Hybridomas

Female BALB/c mouse was purchased from CLEA Japan (Tokyo, Japan). The Animal Care and Use Committee of Tohoku University approved the animal experiments (permit number: 2019NiA-001). The immunization of CD44ec was performed as described previously [29].

The splenic cells were fused with P3U1 cells using polyethylene glycol 1500 (PEG1500; Roche Diagnostics, Indianapolis, IN, USA). The culture supernatants of hybridomas were screened using an enzyme-linked immunosorbent assay (ELISA) against CD44ec. The supernatants were further screened using CHO/CD44v3–10 and parental CHO-K1 cells by flow cytometry using SA3800 Cell Analyzers (Sony Corp. Tokyo, Japan).

C_44_Mab-34 was purified from the cultured supernatants of C_44_Mab-34-producing hybridomas using Ab-Capcher ExTra (ProteNova Co., Ltd., Kagawa, Japan). The purity of C_44_Mab-34 was determined by CBB staining (Appendix A).

### 2.5. ELISA

Fifty-eight synthesized peptides, which cover the CD44v3–10 extracellular domain [30], were synthesized by Sigma-Aldrich Corp (St. Louis, MO, USA). The peptides (1 µg/mL) or CD44ec were immobilized on Nunc Maxisorp 96-well immunoplates (Thermo Fisher Scientific Inc) for 30 min at 37 °C. The immunoplate washing was performed with PBS containing 0.05% (*v*/*v*) Tween 20 (PBST; Nacalai Tesque, Inc.). After the blocking with 1% (*w*/*v*) bovine serum albumin (BSA) in PBST, C_44_Mab-34 (10 µg/mL) was added to each well. Then, the wells were further incubated with anti-mouse immunoglobulins peroxidase-conjugate (1:2000 diluted; Agilent Technologies Inc., Santa Clara, CA, USA). One-Step Ultra TMB (Thermo Fisher Scientific Inc.) was used for enzymatic reactions. An iMark microplate reader (Bio-Rad Laboratories, Inc.) was used to measure the optical density at 655 nm.

### 2.6. Flow Cytometry

CHO/CD44v3–10, CHO-K1, and HSC-3 were harvested using 0.25% trypsin and 1 mM ethylenediamine tetraacetic acid (EDTA; Nacalai Tesque, Inc.). The cells were treated with C_44_Mab-34, C_44_Mab-46, or blocking buffer (control) (0.1% BSA in PBS) for 30 min at 4 °C. Then, the cells were treated with anti-mouse IgG conjugated with Alexa Fluor 488 (1:2000; Cell Signaling Technology, Inc, Danvers, MA, USA) for 30 min at 4 °C. The SA3800 Cell Analyzer and SA3800 software ver. 2.05 (Sony Corporation) were used for fluorescence data collection and analysis, respectively.

### 2.7. Dissociation Constant (K_D_) Determination by Flow Cytometry

Serially diluted C_44_Mab-34 was treated with CHO/CD44v3–10 and HSC-3 cells. Then, the cells were treated with anti-mouse IgG conjugated with Alexa Fluor 488 (1:200). BD FACSLyric and BD FACSuite software version 1.3 (BD Biosciences) were used for fluorescence data collection and analysis, respectively. The GeoMean of each histogram, including primary mAb (C_44_Mab-34) + secondary Ab (Alexa Fluor 488-conjugated anti-mouse IgG) and only secondary Ab (for background), was determined. We further withdrew the background from each data and determined the dissociation constant (*K*_D_) by GraphPad Prism 8 (the fitting binding isotherms to built-in one-site binding models; GraphPad Software, Inc., La Jolla, CA, USA).

### 2.8. Western Blot Analysis

The total cell lysates (10 μg of protein) were denatured by sodium dodecyl sulfate (SDS) sample buffer (Nacalai Tesque, Inc.) in the presence of 2-mercaptoethanol and separated on 7.5% or 5–20% polyacrylamide gels (FUJIFILM Wako Pure Chemical Corporation, Osaka, Japan) and transferred onto polyvinylidene difluoride (PVDF) membranes (Merck KGaA, Darmstadt, Germany). After blocking with 4% skim milk (Nacalai Tesque, Inc.) in PBST, the membranes were incubated with 10 μg/mL of C_44_Mab-34, 10 μg/mL of C_44_Mab-46, 1 μg/mL of NZ-1, or 1 μg/mL of an anti-β-actin mAb (clone AC-15; Sigma-Aldrich Corp.) and then incubated with peroxidase-conjugated anti-mouse immunoglobulins (diluted 1:1000; Agilent Technologies, Inc.) for C_44_Mab-34, C_44_Mab-46, and anti-β-actin. The chemiluminescence signals were obtained with ImmunoStar LD (FUJIFILM Wako Pure Chemical Corporation) and detected using a Sayaca-Imager (DRC Co. Ltd., Tokyo, Japan).

### 2.9. Immunohistochemical Analysis

Formalin-fixed paraffin-embedded (FFPE) sections of the OSCC tissue array (OR601c) were purchased from US Biomax Inc. (Rockville, MD, USA). The OSCC tissue array was autoclaved in EnVision FLEX Target Retrieval Solution High pH (Agilent Technologies, Inc.) for 20 min. After blocking with SuperBlock T20 (Thermo Fisher Scientific, Inc.), the sections were incubated with C_44_Mab-34 (10 μg/mL) and C_44_Mab-46 (1 μg/mL) for 1 h at room temperature and then treated with the EnVision+ Kit for mouse (Agilent Technologies Inc.) for 30 min. The chromogenic reaction was conducted using 3,3′-diaminobenzidine tetrahydrochloride (DAB; Agilent Technologies Inc.). The counterstaining was performed using hematoxylin (FUJIFILM Wako Pure Chemical Corporation). To examine the sections and obtain images, we used a Leica DMD108 (Leica Microsystems GmbH, Wetzlar, Germany).

## 3. Results

### 3.1. Development of an Anti-CD44v7/8 mAb, C_44_Mab-34

In this study, we purified human CD44ec as an immunogen (Figure 1). One mouse was immunized with CD44ec, and hybridomas were seeded into 96-well plates. The supernatants were first screened by the reactivity to CD44ec by ELISA. Subsequently, the supernatants, which were positive for CHO/CD44v3–10 cells and negative for CHO-K1 cells, were further selected using flow cytometry. Finally, anti-CD44 mAb-producing clones were established by limiting dilution. Among them, C_44_Mab-34 (IgG_1_, kappa) was shown to recognize CD44p421–440 (GHQAGRRMDMDSSHSTTLQP), which corresponds to the variant 7- and variant 8-encoded sequence (Appendix A). In contrast, C_44_Mab-34 never recognized other CD44v3–10 extracellular regions. These results indicate that C_44_Mab-34 specifically recognizes the border region between variants 7 and 8.

### 3.2. Flow Cytometric Analysis of C_44_Mab-34 to CD44-Expressing Cells

We next investigated the reactivity of C_44_Mab-34 against CHO/CD44v3–10 and CHO/CD44s cells by flow cytometry. C_44_Mab-34 recognized CHO/CD44v3–10cells in a dose-dependent manner (Figure 2A) but not CHO/CD44s (Figure 2B) or CHO-K1 (Figure 2C) cells. The CHO/CD44 cells were recognized by an anti-pan-CD44 mAb, C_44_Mab-46 [29] (Appendix A). C_44_Mab-34 also recognized the OSCC cell line HSC-3 (Figure 2D) in a dose-dependent manner.

We next determined the binding affinity of C_44_Mab-34 with CHO/CD44v3–10 and HSC-3 using flow cytometry. The *K*_D_ of CHO/CD44v3–10 and HSC-3 was 1.4 × 10^−9^ and 3.2 × 10^−9^ M, respectively. These results indicate that C_44_Mab-34 possesses a moderate affinity for CD44v3–10-expressing cells (Figure 3).

### 3.3. Western Blot Analysis

We next performed Western blot analysis to assess the sensitivity of C_44_Mab-34. As shown in Figure 4A, an anti-pan-CD44 mAb, C_44_Mab-46, recognized the lysates from both CHO/CD44s (75~100 kDa) and CHO/CD44v3–10 (>180 kDa). C_44_Mab-34 mainly detected CD44v3–10 as more than 180-kDa bands. However, C_44_Mab-34 did not detect any bands from the lysates of CHO-K1 and CHO/CD44s cells (Figure 4B). These results indicate that C_44_Mab-34 can detect CD44v3–10.

### 3.4. Immunohistochemical Analysis against Tumor Tissues Using C_44_Mab-34

We next examined whether C_44_Mab-34 could be used for immunohistochemical analyses using FFPE sections. We used sequential sections of an OSCC tissue microarray. In a well-differentiated OSCC section, the clear membranous staining in OSCC was observed by C_44_Mab-34 and C_44_Mab-46 (Figure 5A,B). In an OSCC section with the stromal-invaded phenotype, C_44_Mab-34 strongly stained stromal-invaded OSCC and could clearly distinguish tumor cells from stromal tissues (Figure 5C). In contrast, C_44_Mab-46 stained both invaded tumor cells and surrounding stroma cells (Figure 5D). In Figure 5E,F, C_44_Mab-34 and C_44_Mab-46 never stained tumor tissue, but clear stromal staining was observed by C_44_Mab-46 (Figure 5F). We have summarized the data of the immunohistochemical analysis of CD44 expression in tumor cells in Table 1; C_44_Mab-34 stained 42 out of 49 (86%) cases of OSCC. These results indicate that C_44_Mab-34 is useful for the immunohistochemical analysis of FFPE tumor sections.

## 4. Discussion

Head and neck cancer is the seventh most common type of cancer worldwide, and it exhibits aggressive development in clinical settings [58]. Head and neck cancer remains a complex disease with a profound impact on patients and their quality of life after surgical ablation and therapies. Knowledge of the disease has been accumulated with regard to tumor biology and prevention, and therapeutic options have been simultaneously developed [58]. HNSCC is the most common type of head and neck cancer, and it has been revealed as the second-highest CD44-expressing cancer type in the Pan-Cancer Atlas [59]. CD44 overexpression is associated with poor prognosis and resistance to therapy [60,61,62]. Reduced CD44 expression leads to the growth suppression of tumor cells [17,63]. Therefore, CD44 is considered an important target for mAb therapies. In this study, we developed a novel anti-CD44v7/8 mAb, C_44_Mab-34, and showed multiple applications to flow cytometry (Figure 2 and Figure 3), Western blotting (Figure 4), and the immunohistochemistry of OSCC (Figure 5).

An anti-CD44v7/8 mAb (clone VFF-17) was previously developed, and it has been mainly used for the immunohistochemistry of normal tissue and tumors [64,65]. The epitope of VFF-17 mAb was determined by binding studies with fusion proteins encoding v7 or v8 exons, either alone or in combination [66]. However, a detailed amino acid sequence of the epitope has not been determined. As shown in Appendix A, C_44_Mab-34 recognizes CD44p421–440 [GHQAGRRMD (included in v7) + MDSSHSTTLQP (included in v8)]. In contrast, C_44_Mab-34 has never recognized CD44p411–430 (FNPISHPMGRGHQAGRRMD (included in v7) + M (included in v8)) or CD44p431–450 (DSSHSTTLQPTANPNTGLVE (included in v8)). These results suggest that C_44_Mab-34 recognizes the border sequence between v7 and v8. In addition, CD44 is known to be heavily glycosylated [67], and the glycosylation pattern is thought to depend on the host cells. Since the epitope of C_44_Mab-34 contains predicted and confirmed *O*-glycan sites [67], further studies are needed on whether the recognition of C_44_Mab-34 is affected by glycosylation.

Among many CD44v types, CD44v8–10, CD44v6–10, CD44v4–10, and CD44v3–10 were mainly detected in SCC cells by semi-quantitative RT-PCR analysis (manuscript submitted). Since C_44_Mab-34 recognizes the border sequence between v7 and v8 (Appendix A), C_44_Mab-34 can distinguish CD44v8–10 and the longer CD44v (v6-10, v4-10, and v3–10). Furthermore, the inclusion of these variants (from v8-10 to the longer variants) is promoted by EGF signaling [68,69]. If the expression of CD44v8-10 and the longer variants are differently regulated in normal and tumor cells, C_44_Mab-34 could contribute to tumor diagnosis and therapy. We are now investigating the C_44_Mab-34 reactivity against other tumor tissues together with the epitope analyses.

∆Np63 is known as a marker of basal cells of stratified epithelium and SCC [70]. ∆Np63 mediates HA metabolism and signaling [71]. Specifically, ΔNp63 directly binds to the p63-binding sequence on the promoter/enhancer region of the CD44 gene [71]. In whole-exome sequencing data analysis from 74 HNSCC–normal pairs, the ∆Np63-encoded gene, *TP63*, was identified as a significantly mutated gene that results in the activation of the ∆Np63 pathway [72]. The relationship between ∆Np63 activation and CD44 transcription should be investigated in future studies. Furthermore, the mechanism of the variant 7/8 inclusion by alternative splicing remains to be determined.

An anti-pan CD44 mAb, RG7356, demonstrated some efficacy and an acceptable safety profile in the phase I study. However, the study was terminated due to no evidence of a clinical and dose–response relationship with RG7356 [73]. Furthermore, a variant 6-specific CD44 mAb-drug conjugate (bivatuzumab–mertansine) was also evaluated in clinical trials. However, lethal epidermal necrolysis halted further development. The efficient accumulation of mertansine in the skin was most likely responsible for the high toxicity [74,75]. Therefore, the therapeutic effects of CD44 mAbs have been disappointing until now.

Near-infrared photoimmunotherapy (NIR-PIT) is a novel tumor therapy that uses a targeted mAb–photoabsorber conjugate (APC) [76]. The mAb binds to the targeted cell surface antigen, and the photoactivatable dye IRDye700DX (IR700) induces the disruption of the cellular membrane after NIR-light exposure. Since NIR-light exposure can be performed at tumor sites locally, APC can exert antitumor effect selectivity while minimizing damage to the surrounding tissue [77,78]. Preclinical studies indicate that NIR-PIT induces tumor necrosis and immunogenic cell death through the induction of innate and adaptive immunity [79]. A first-in-human phase I and II trial of NIR-PIT with RM-1929 (an anti-epidermal growth factor receptor mAb, cetuximab–IR700 conjugate) in patients with inoperable HNSCC was conducted and exhibited the efficacy [80].

A preclinical study of anti-CD44 mAb-based NIR-PIT has been reported [81]. The study used an anti-mouse/human pan-CD44 mAb, IM7, conjugated with IR700 (CD44–IR700) in a syngeneic mouse model of OSCC. The CD44–IR700 can induce significant antitumor responses after a single injection of the conjugate and NIR-light exposure in CD44-expressing OSCC tumors [81]. As shown in Figure 5D,F, a pan-CD44 mAb, C_44_Mab-46, recognized not only tumor cells but also stromal tissue and probably immune cells, which are important for antitumor immunity. Therefore, CD44v is a promising tumor antigen for NIR-PIT, which could be a new modality for OSCC with locoregional recurrence.

We have previously produced recombinant antibodies that are converted to a mouse IgG_2a_ subclass from mouse IgG_1_. Furthermore, we produced defucosylated IgG_2a_ mAbs using fucosyltransferase 8-deficient CHO-K1 cells to potentiate antibody-dependent cellular cytotoxicity. The defucosylated mAbs showed potent antitumor activity in mouse xenograft models [33,82,83,84,85,86,87,88]. Therefore, a class-switched and defucosylated version of C_44_Mab-34 is required to evaluate the antitumor activity in vivo.

## Figures and Tables

**Figure 1 biomedicines-11-01099-f001:**
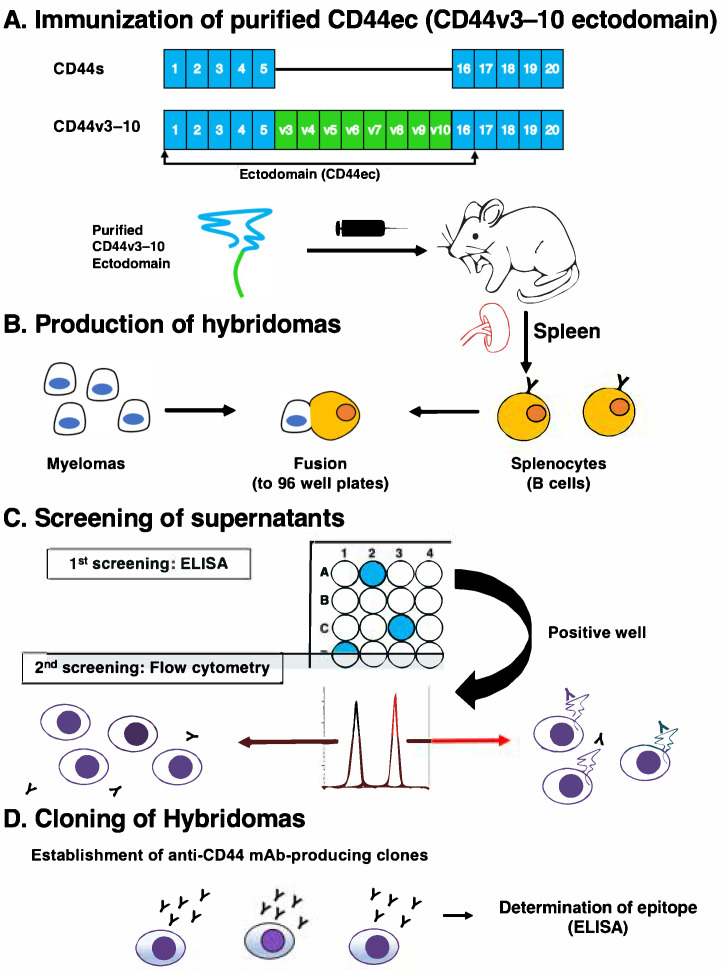
A schematic illustration of anti-human CD44 mAb production. (**A**) Purified CD44v3–10 ectodomain was intraperitoneally injected into BALB/c mouse. (**B**) Hybridomas were produced by fusion of the splenocytes and P3U1 cells. (**C**) The screening was performed by enzyme-linked immunosorbent assay (ELISA) and flow cytometry using parental CHO-K1 and CHO/CD44v3–10cells. (**D**) After cloning and additional screening, a clone C_44_Mab-34 (IgG_1_, kappa) was established. Furthermore, the binding epitopes were determined by ELISA using peptides, which cover the extracellular domain of CD44v3–10.

**Figure 2 biomedicines-11-01099-f002:**
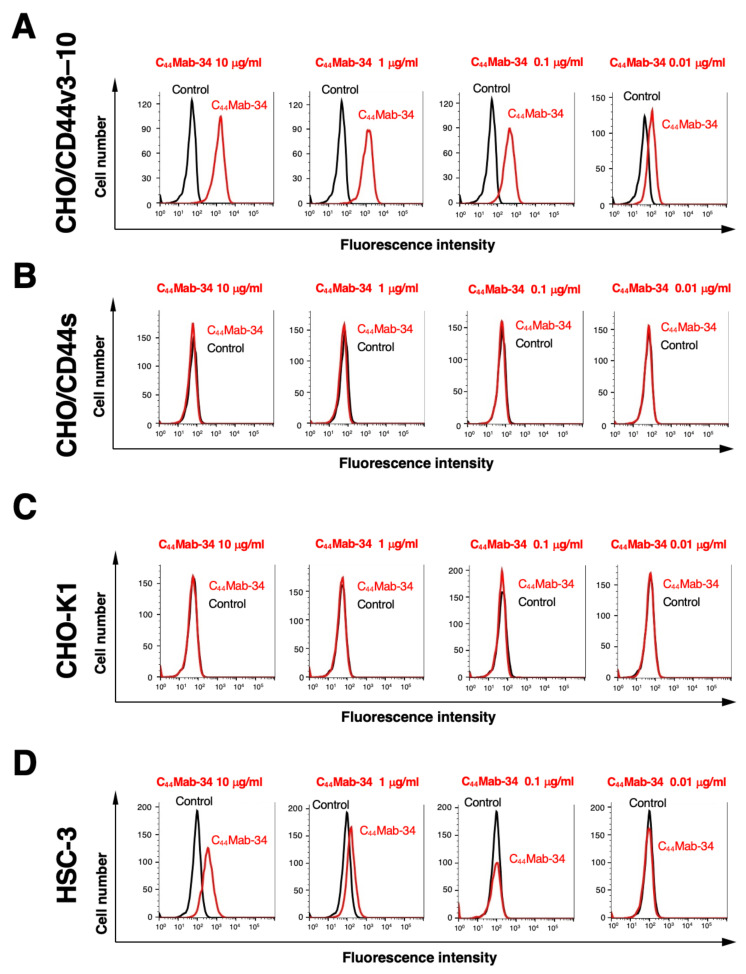
Flow cytometry using C_44_Mab-34 against CD44-expressing cells. CHO/CD44v3–10 (**A**), CHO/CD44s (**B**), CHO-K1 (**C**), and HSC-3 (**D**) cells were treated with C_44_Mab-34, followed by treatment with anti-mouse IgG conjugated with Alexa Fluor 488 (red line). The black line represents the negative control (blocking buffer).

**Figure 3 biomedicines-11-01099-f003:**
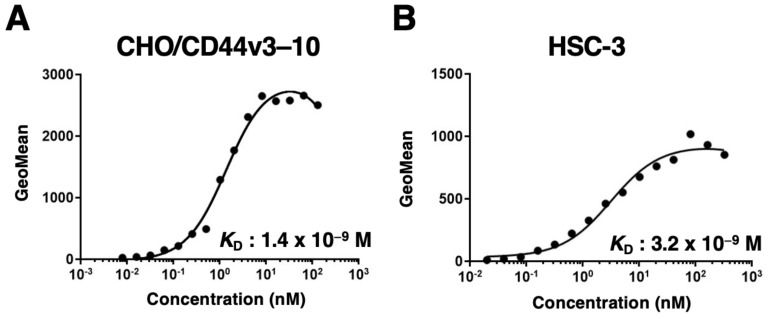
The binding affinity of C_44_Mab-34 to CD44-expressing cells. CHO/CD44v3–10 (**A**) and HSC-3 (**B**) cells were suspended in serially diluted C_44_Mab-34 at indicated concentrations. Then, cells were treated with anti-mouse IgG conjugated with Alexa Fluor 488. Fluorescence data were collected, followed by the calculation of the dissociation constant (*K*_D_) by GraphPad PRISM 8.

**Figure 4 biomedicines-11-01099-f004:**
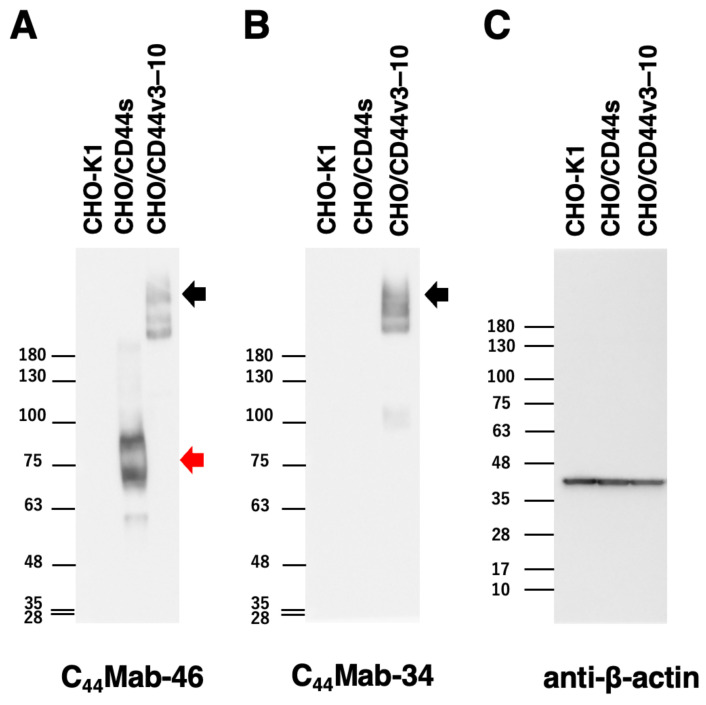
Western blot analysis using C_44_Mab-34. The cell lysates from CHO-K1, CHO/CD44s, and CHO/CD44v3–10 (10 µg) were electrophoresed and transferred onto polyvinylidene fluoride (PVDF) membranes. The membranes were incubated with 10 µg/mL of C_44_Mab-46 (**A**), 10 µg/mL of C_44_Mab-34 (**B**), and 1 µg/mL of an anti-β-actin mAb (**C**). Then, the membranes were incubated with anti-mouse immunoglobulins conjugated with peroxidase for C_44_Mab-46, C_44_Mab-34, and an anti-β-actin mAb. The red arrows indicate the CD44s (75~100 kDa). The black arrows indicate the CD44v3–10.

**Figure 5 biomedicines-11-01099-f005:**
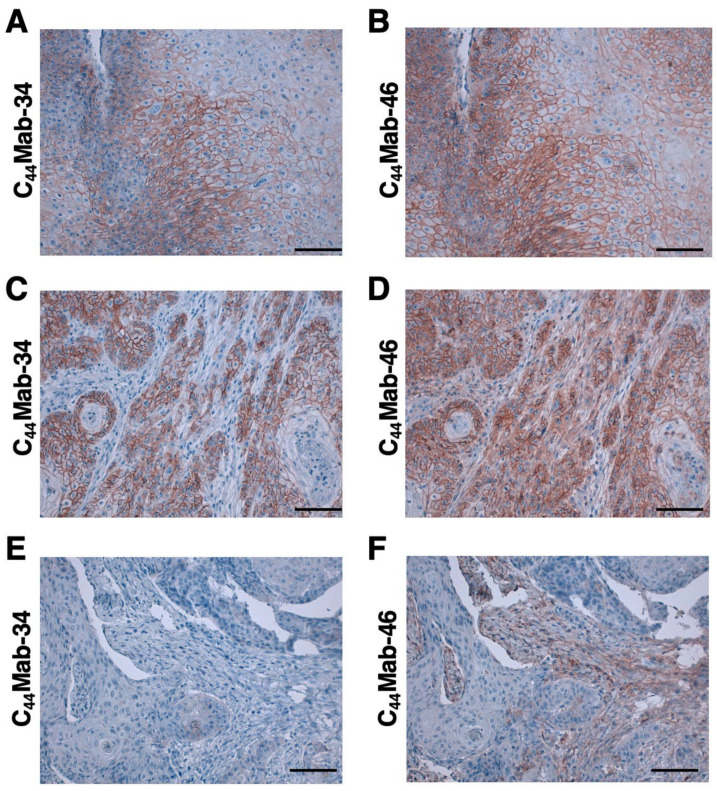
Immunohistochemical analysis using C_44_Mab-34 and C_44_Mab-46 against OSCC tissues. After antigen retrieval, serial sections of OSCC tissue array (Catalog number: OR601c) were incubated with 10 µg/mL of C_44_Mab-34 (**A, C, E**) or 1 µg/mL of C_44_Mab-46 (**B, D, F**), followed by treatment with the Envision+ kit. The chromogenic reaction was conducted using 3,3′-diaminobenzidine tetrahydrochloride (DAB). The counterstaining was performed using hematoxylin. Scale bar = 100 µm.

**Table 1 biomedicines-11-01099-t001:** Immunohistochemical analysis using C_44_Mab-34 and C_44_Mab-46 against OSCC.

No	Age	Sex	Organ/Anatomic Site	Pathology Diagnosis	TNM	C_44_Mab-34	C_44_Mab-46
1	78	M	Tongue	Squamous cell carcinoma of tongue	T2N0M0	+	+
2	40	M	Tongue	Squamous cell carcinoma of tongue	T2N0M0	+	++
3	35	F	Tongue	Squamous cell carcinoma of tongue	T2N0M0	+++	++
4	61	M	Tongue	Squamous cell carcinoma of tongue	T2N0M0	+++	+++
5	41	F	Tongue	Squamous cell carcinoma of tongue	T2N0M0	+	+
6	64	M	Tongue	Squamous cell carcinoma of right side of tongue	T2N2M0	+	++
7	76	M	Tongue	Squamous cell carcinoma of tongue	T1N0M0	+	++
8	50	F	Tongue	Squamous cell carcinoma of tongue	T2N0M0	+++	++
9	44	M	Tongue	Squamous cell carcinoma of tongue	T2N1M0	+++	++
10	53	F	Tongue	Squamous cell carcinoma of tongue	T1N0M0	+	++
11	46	F	Tongue	Squamous cell carcinoma of tongue	T2N0M0	-	+
12	50	M	Tongue	Squamous cell carcinoma of root of tongue	T3N1M0	+++	+
13	36	F	Tongue	Squamous cell carcinoma of tongue	T1N0M0	+++	+++
14	63	F	Tongue	Squamous cell carcinoma of tongue	T1N0M0	+	+
15	46	M	Tongue	Squamous cell carcinoma of tongue	T2N0M0	++	-
16	58	M	Tongue	Squamous cell carcinoma of tongue	T2N0M0	+	+
17	64	M	Lip	Squamous cell carcinoma of lower lip	T1N0M0	+++	+++
18	57	M	Lip	Squamous cell carcinoma of lower lip	T2N0M0	++	+++
19	61	M	Lip	Squamous cell carcinoma of lower lip	T1N0M0	+++	++
20	60	M	Gum	Squamous cell carcinoma of gum	T3N0M0	+	+
21	60	M	Gum	Squamous cell carcinoma of gum	T1N0M0	+++	+++
22	69	M	Gum	Squamous cell carcinoma of upper gum	T3N0M0	+	++
23	53	M	Bucca cavioris	Squamous cell carcinoma of bucca cavioris	T2N0M0	++	+
24	55	M	Bucca cavioris	Squamous cell carcinoma of bucca cavioris	T1N0M0	++	+
25	58	M	Tongue	Squamous cell carcinoma of base of tongue	T1N0M0	+	++
26	63	M	Oral cavity	Squamous cell carcinoma	T1N0M0	++	++
27	48	F	Tongue	Squamous cell carcinoma of tongue	T1N0M0	++	+
28	80	M	Lip	Squamous cell carcinoma of lower lip	T1N0M0	+++	+++
29	77	M	Tongue	Squamous cell carcinoma of base of tongue	T2N0M0	+++	++
30	59	M	Tongue	Squamous cell carcinoma of tongue	T2N0M0	++	-
31	77	F	Tongue	Squamous cell carcinoma of tongue	T1N0M0	+	++
32	56	M	Tongue	Squamous cell carcinoma of root of tongue	T2N1M0	+	+
33	60	M	Tongue	Squamous cell carcinoma of tongue	T2N1M0	+	++
34	62	M	Tongue	Squamous cell carcinoma of tongue	T2N0M0	+++	++
35	67	F	Tongue	Squamous cell carcinoma of tongue	T2N0M0	+++	++
36	47	F	Tongue	Squamous cell carcinoma of tongue	T2N0M0	+++	+++
37	37	M	Tongue	Squamous cell carcinoma of tongue	T2N1M0	-	-
38	55	F	Tongue	Squamous cell carcinoma of tongue	T2N0M0	++	++
39	56	F	Bucca cavioris	Squamous cell carcinoma of bucca cavioris	T2N0M0	++	+
40	49	M	Bucca cavioris	Squamous cell carcinoma of bucca cavioris	T1N0M0	-	-
41	45	M	Bucca cavioris	Squamous cell carcinoma of bucca cavioris	T2N0M0	-	-
42	42	M	Bucca cavioris	Squamous cell carcinoma of bucca cavioris	T3N0M0	++	++
43	44	M	Jaw	Squamous cell carcinoma of right drop jaw	T1N0M0	+	+++
44	40	F	Tongue	Squamous cell carcinoma of base of tongue	T2N0M0	-	++
45	49	M	Bucca cavioris	Squamous cell carcinoma of bucca cavioris	T1N0M0	+++	+++
46	56	F	Tongue	Squamous cell carcinoma of base of tongue	T2N0M0	-	-
47	42	M	Bucca cavioris	Squamous cell carcinoma of bucca cavioris	T3N0M0	+++	+++
48	87	F	Face	Squamous cell carcinoma of left side of face	T2N0M0	+	+
49	50	M	Gum	Squamous cell carcinoma of gum	T2N0M0	-	-

## Data Availability

The data presented in this study are available in the article and Appendix A.

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
