# Peer review of "Development of a Novel Anti-CD44 Variant 7/8 Monoclonal Antibody, C44Mab-34, for Multiple Applications against Oral Carcinomas"

_biomedicines, 2023, doi:10.3390/biomedicines11041099_

Round 1

Reviewer 1 Report

This is a scientifically sound study describing the discovery of a Mab that could be used in diagnostics, characterization and perhaps as a therapeutic for HNSCC. The Mab discovered in this study is specific to the 7/8 variant among the CD44v expressed in HNSCC. Approaches used for validating this Mab are well established and the authors have identified other Mabs that have showed utility for OSCC.

1. For CD44Mab-34 flow cytometry experiments, can you comment on the comparison to CD44Mab-44 results shown in the supplement? How is this Mab superior?

2. Table 1  - Can you comment on the comparison in expression levels seen for 34 and 46 and whether there are any patterns that are expected or unexpected given the epitopes that the two antibodies recognize.

3. In the discussion section can you comment on the specific utility of 34 compared to the other CD44 Mabs already being studied for therapeutic purposes?

Author Response

This is a scientifically sound study describing the discovery of a Mab that could be used in diagnostics, characterization and perhaps as a therapeutic for HNSCC. The Mab discovered in this study is specific to the 7/8 variant among the CD44v expressed in HNSCC. Approaches used for validating this Mab are well established and the authors have identified other Mabs that have showed utility for OSCC.

  1. For CD44Mab-34 flow cytometry experiments, can you comment on the comparison to CD44Mab-46 results shown in the supplement? How is this Mab superior?

The reactivity of C44Mab-34 to CD44v3-10 cells (Figure 2) is superior to C44Mab-46 (Supplementary Figure S2) at the low concentration (0.01 µg/ml).

Furthermore, in the binding affinity (KD value) to CD44 expressing cells, C44Mab-34 exhibited the 10−9 order as shown in Figure 3. In contrast, C44Mab-46 exhibited the 10−8 order (ref 29). Therefore, C44Mab-34 is superior to C44Mab-46 in the binding affinity.

  1. Table 1 - Can you comment on the comparison in expression levels seen for 34 and 46 and whether there are any patterns that are expected or unexpected given the epitopes that the two antibodies recognize.

As shown in Figure 5, membranous staining patterns of C44Mab-34 and C44Mab-46 were similar in OSCC. To acheave the staining, we used 10 µg/mL of C44Mab-34 and 1 µg/mL of C44Mab-46.

C44Mab-46 positive signal includes both CD44s and all CD44v. In contrast, C44Mab-34 positive signal includes only CD44 containing variant 7 and 8 (CD44v7/8). Therefore, the abundance of CD44v7/8 is thought to be low.

  1. In the discussion section can you comment on the specific utility of 34 compared to the other CD44 Mabs already being studied for therapeutic purposes?

We added the possibility of the specific utility of C44Mab-34 in discussion, as follows.

Among CD44v, CD44v8–10, CD44v6–10, CD44v4–10, and CD44v3–10 were mainly detected in SCC cells by semi-quantitative RT-PCR analysis (manuscript submitted). Since C44Mab-34 recognizes the border sequence between v7 and v8 (supplemental Table 1), C44Mab-34 can distinguish CD44v8–10 and the longer CD44v (v6–10, v4–10, and v3–10). Furthermore, these variants inclusion (from v8–10 to the longer variants) is promoted by EGF signaling [68,69]. If the expression of CD44v8–10 and the longer variants are differently regulated in normal and tumor cells, C44Mab-34 could contribute the tumor diagnosis and therapy. We are investigating the reactivity to other tumor tissue together with the epitope analysis.

Reviewer 2 Report

This article aims to develop a monoclonal antibody, C44Mab-34 that targets CD44 variant 7/8. To achieve this aim, the authors produced C44Mab-34 mAbs and then tested its binding affinity, specificity, and immunohistochemistry. Overall, this is a very interesting article and has many meaningful applications, such as on oral carcinomas. However, the article needs more supportive information for the current results.

1) In section 3.1, the authors used CHO/CD44v3-10 cells as a selection for anti-CD44 mAb. How do authors ensure the quantity of CHO cells that can express CD44v3-10? It is likely the CHO cells do not express CD44v3-10 very well or these cells possibly express other protein variants. In the latter case, C44Mab-34 might bind to non-CD44 variants, leading to the results in section 3.2 being falsely positive.

2) Please show the purity for CD44v3-10 and more importantly for the anti-CD44 mAb (e.g., using HPLC data) in the supplementary information.

3) Please show the cell gating for the flow cytometric results in Figure 2 to ensure the cells of analysis were selected properly.

4) In Figure 3, the Kd calculation would be more accurate, if the authors treated CHO/CD44v3-10 and HSC-3 cells with AF488 IgG and use the corresponding results to move the background of current your results. Because it is likely that AF488 IgG binds to the cells nonspecifically, leading your Kd to be inaccurate.

5) In Figure 4B, the CHO/CD44s column has CD44v3-10 bands, while CHO/CD44v3-10 column has no such bands; the authors should give an explanation.

6) In theory, results in Figure 4 cannot give us the conclusion shown in line 252-253 (authors stating ‘These results indicated that 252 C44Mab-34 specifically detects CD44v3–10’). Because the authors tested the binding of your Mab to different cell lysates, which contain millions of different proteins. Among these proteins, there are some proteins with molecular weight (MW) similar to MW of CD44s or to MW of CD44v3-10. It would be better if the authors can provide the binding results of the Mab to the purified CD44v-10 and CD44s proteins. Another option is to use a verified Mab as a positive control.

7) DeltaNp63, RG7356, NIR-PIT, and IgG2a in the discussion are not directly related to your antibody development. Please discuss more to compare your antibody with its potential competitors in the field and show the significance of your antibody.

8) At line 217, please change “ant-human CD44” to “anti-human CD44”.

Round 2

Reviewer 1 Report

I do not have any more comments